# DeepBrainPrint: A Novel Contrastive Framework for Brain MRI Re-Identification

**Lemuel Puglisi**[1]                                          LEMUEL.PUGLISI@QUEENSQUAREANALYTICS.COM
**for the Alzheimer's Disease Neuroimaging Initiative**
**Frederik Barkhof**[1,2,3,5]                                              F.BARKHOF@UCL.AC.UK
**Daniel C. Alexander**[1,2]                                           D.ALEXANDER@UCL.AC.UK
**Geoffrey JM Parker**[1,2,5]                                         GEOFF.PARKER@UCL.AC.UK
**Arman Eshaghi**[1,2,5]                                               A.ESHAGHI@UCL.AC.UK
**Daniele Ravì**[1,2,4]                                                  D.RAVI@UCL.AC.UK

[1] *Queen Square Analytics, London, UK*

[2] *Centre for Medical Image Computing (CMIC), University College London, London, UK*

[3] *VU University Medical Center, Amsterdam, the Netherlands*

[4] *School of Physics, Engineering and Computer Science University of Hertfordshire, Hatfield, UK*

[5] *NMR Unit, Queen Square Multiple Sclerosis Centre, Department of Neuroinflammation, Queen Square Institutes of Neurology, Faculty of Brain Sciences, University College London, London, UK*

**Editors:** Accepted for publication at MIDL 2023

## Abstract

Recent advances in MRI have led to the creation of large datasets. With the increase in data volume, it has become difficult to locate previous scans of the same patient within these datasets (a process known as re-identification). To address this issue, we propose an AI-powered medical imaging retrieval framework called DeepBrainPrint, which is designed to retrieve brain MRI scans of the same patient. Our framework is a semi-self-supervised contrastive deep learning approach with three main innovations. First, we use a combination of self-supervised and supervised paradigms to create an effective brain fingerprint from MRI scans that can be used for real-time image retrieval. Second, we use a special weighting function to guide the training and improve model convergence. Third, we introduce new imaging transformations to improve retrieval robustness in the presence of intensity variations (i.e. different scan contrasts), and to account for age and disease progression in patients. We tested DeepBrainPrint on a large dataset of T1-weighted brain MRIs from the Alzheimer's Disease Neuroimaging Initiative (ADNI) and on a synthetic dataset designed to evaluate retrieval performance with different image modalities. Our results show that DeepBrainPrint outperforms previous methods, including simple similarity metrics and more advanced contrastive deep learning frameworks.

**Keywords:** Brain MRI Fingerprint, Re-Identification, Deep metric Learning

## 1. Introduction

The demand for radiologists to interpret brain MRIs is increasing due to the growing number of MRI scans and advances in MRI technology. To alleviate the burden on hospitals, tools for efficient image retrieval can be crucial. For example, retrieving former scans from the same patient can provide valuable insights for analyzing their disease progression over time. Metadata, such as anonymized IDs, could be useful for this purpose, but their association

Puglisi[1] Initiative Barkhof[1,2,3,5] Alexander[1,2] Parker[1,2,5] Eshaghi[1,2,5] Ravì[1,2,4]

with each individual may not always be accurate. In fact, different IDs may be assigned to the same subject within the same hospital or across different hospitals. To address this issue, our work focuses on efficient subject re-identification by extracting a unique fingerprint from a subject's brain MRI.

While several solutions exist for this purpose, all of them have limitations, mainly regarding efficiency and generalization to different modalities.

Facial recognition algorithms have been proposed to address subject re-identification using MRIs (Schwarz et al., 2019). However, faces and skulls are often removed during the anonymization process, making this solution impractical.

Other approaches involve aligning MRIs into a common template and comparing them using image similarity based on intensity values (e.g., Minimum Cross Entropy or Structural Similarity Index (Al-bory and Elsheh, 2018)). However, the number of voxels is often too large to efficiently evaluate image similarity. Additionally, each voxel does not carry anatomic information that characterizes the brain, leading to the retrieval of images that do not necessarily represent the same subject (Faria et al., 2015). To overcome this limitation, Wachinger et al. (Wachinger et al., 2015) proposed a solution that captures geometric information by using the spectrum of the Laplace-Beltrami operator on meshes derived from brain MRIs, making the approach robust to intensity variations. Nevertheless, the requirement of brain segmentation hinders the efficiency of this approach. Another recent approach for brain fingerprinting is proposed in (Chauvin et al., 2020), which uses 3D SIFT-Rank descriptors to match brain MRIs. Nonetheless, this approach faces difficulty in generalizing to different modalities. One possible solution to the modality issue is to utilize a multi-modal approach, as demonstrated in the work of Kumar et al. (Kumar et al., 2018). However, this approach assumes the availability of all required modalities, which cannot always be guaranteed.

Recent advancements in deep metric learning offer a promising and unexplored solution for enhancing the efficiency and generalization of brain fingerprinting and re-identification. These approaches develop efficient representation functions that map each scan into an embedding space, ensuring that the distances in the new manifold reflect the initial similarity between scans. Approaches based on deep metric learning can be divided into two main categories: fully-supervised and self-supervised. Fully-supervised methods rely on patient identification (ID) to extract information from MRI scans of the same subject taken at different times (follow-up scans). They ensure that representations of scans from the same patient are similar while keeping representations of scans from different subjects distinct. Self-supervised methods, on the other hand, generate their own representations of the data by solving a task derived from the input data, without relying on patient ID. They can be trained even with cross-sectional datasets or when patient IDs are not available.

In the fully-supervised category, we found deep Siamese networks (Chopra et al., 2005) which use labelled data to train a model using a contrastive loss (Kumar BG et al., 2016) such as Triplet (Hoffer and Ailon, 2015), SoftTriple (Qian et al., 2019), Proxy-NCA (Movshovitz-Attias et al., 2017), and InfoNCE (Oord et al., 2018)). However, recent research (Musgrave et al., 2020) has shown that these supervised contrastive solutions have reached a limit in terms of improvements.

Most of the self-supervised approaches available today rely on image transformations to create multiple distorted images that belong to the same subject (e.g., BarlowTwins (Zbon-

tar et al., 2021), SimCLR (Chen et al., 2020)). While these self-supervised approaches have demonstrated impressive results on natural images, the transformations they use may not be suitable for MRIs. Additionally, avoiding completely the usage of available labels (i.e., subject IDs) during model training could be a limitation in our context, as labels could be exploited to improve model performance.

In this work, we propose a hybrid solution that aims to overcome the limitations above. Specifically, we extend the approach in (Zbontar et al., 2021) by adding three main novelties: i) we combine a self-supervised and a supervised paradigm to optimize a multi-task problem that leverages available labels and improves retrieval performance; ii) we introduce a method to weight the importance of two training tasks considered in our pipeline and improve model convergence; iii) we propose a new set of image transformations designed specifically for brain MRIs that are necessary to make the framework robust to scan variability. Our solution operates on a single 2D slice, facilitating real-time retrieval.

To the best of our knowledge, this is the first semi-self-supervised approach for brain MRI re-identification. Furthermore, our domain-specific image transformations are designed to make the retrieval robust to intensity changes, disease progression, and patient ageing, leading the framework to learn an adequate representation of brain morphology and achieving a recall performance close to 99% during retrievals.

## 2. Method

In this section, we provide the details of our framework which include i) scan pre-processing, ii) training pipeline, iii) image transformations, and iv) training settings.

### 2.1. Pre-processing

This block aims to reduce critical but non-informative variations in the data and prepare scans for model training. Four sequential steps are used to process each scan: i) co-linear registration to an MNI152 template (using the ANTs library), ii) removal of 5% of intensity outliers from the entire scan, iii) normalization of intensity using the z-score on each voxel, and iv) extraction of the central slice $x$ from the axial plane.

### 2.2. Proposed DeepBrainPrint pipeline

The blocks of our training pipeline are depicted in Figure 1. The pipeline is divided into two main branches aimed to optimize two different properties of the embedding vectors: i) the self-supervised branch (in light green) ensures that all the representations obtained from scans from the same subject are similar to each other even if they are affected by some variability (e.g., different contrasts, different scanners, different protocols, etc.) and ii) the supervised contrastive branch (in light orange) ensures that representations obtained from follow-up scans from the same subject are similar to each other whereas scans from different subjects are dissimilar from each other. In particular, the self-supervised branch is inspired by the work proposed in (Zbontar et al., 2021) in which we integrate new domain-specific image transformations designed for brain MRI (see Section 2.3). The supervised branch, instead, follows a standard contrastive setting with triplets used to optimize a loss inspired by (Oord et al., 2018).

PUGLISI[1] INITIATIVE BARKHOF[1,2,3,5] ALEXANDER[1,2] PARKER[1,2,5] ESHAGHI[1,2,5] RAVÌ[1,2,4]

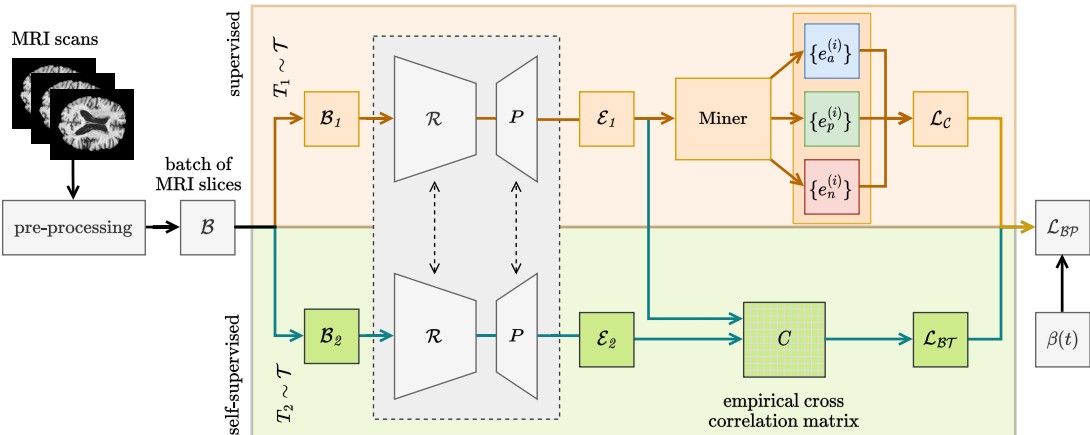

Figure 1: Workflow of our DeepBrainPrint pipeline which is divided into two main branches: a fully supervised branch (in orange) and a self-supervised branch (in green).

The training starts by sampling random batches $\mathcal{B}$ with size $b$ containing a set of pre-processed slices $x_1, \ldots, x_b$. Then, for each 2D slice $x_i \in \mathcal{B}, i = 1, \ldots, b$ we sample two transformations $T_1, T_2 \sim \mathcal{T}$ (see Section 2.3) and obtain two distorted version $x_i' = T_1(x_i)$ and $x_i'' = T_2(x_i)$ of the scan $x_i$. By repeating this process for each slice $x_i$, we generate two sets of batches defined as:

$$\mathcal{B}_1 = \{x_i' = T_1(x) \mid x_i \in \mathcal{B}, T_1 \sim \mathcal{T}\}$$
$$\mathcal{B}_2 = \{x_i'' = T_2(x) \mid x_i \in \mathcal{B}, T_2 \sim \mathcal{T}\} \tag{1}$$

These two batches are then passed to the encoder network $\mathcal{R}$, which shares weights between the two branches of the pipeline using a Siamese configuration. $\mathcal{R}$ maps each transformed slice to a low-dimensional vector (the final "representation") having size $z$. Following this, a projection head $P$, having also shared weights, maps the representations obtained from $\mathcal{R}$ in vectors lying in a non-linear space with dimension $s$. We will refer to these new vectors as the "embeddings" $e$. Let's define the composition of the encoder and the projection head as $\mathcal{M}(\cdot) = P(\mathcal{R}(\cdot))$ and the corresponding embedding batches as $\mathcal{E}_1 = \mathcal{M}(\mathcal{B}_1)$ and $\mathcal{E}_2 = \mathcal{M}(\mathcal{B}_2)$. The pipeline proceeds by processing $\mathcal{E}_1$ and $\mathcal{E}_2$ and computing two main losses.

The self-supervised branch computes the loss $\mathcal{L}_{\mathcal{BT}}$ inspired by (Zbontar et al., 2021) and defined as follows:

$$\mathcal{L}_{\mathcal{BT}} = \sum_i (1 - c_{ii})^2 + \lambda \sum_i \sum_{j \neq i} c_{ij}^2; \tag{2}$$

where $c_{ij}$ is an element of the cross-correlation matrix $C$, computed on $(\mathcal{E}_1, \mathcal{E}_2)$. Each $c_{ij}$ indicates how much the $i$-th feature in the embedding vector is correlated to the $j$-th feature. The rightmost term called the "redundancy reduction term" ensures that the correlation between features is minimized, and the hyperparameter $\lambda$ weights its final contribution in the loss. Instead, the left term, called the "invariance term", ensures that features extracted from different transformations of the same image are correlated, which makes the embedded representation invariant to the proposed image transformations.

On the other side, the supervised branch, which is trained simultaneously with the other branch, optimizes the loss $\mathcal{L}_\mathcal{C}$ inspired by (Oord et al., 2018) and defined as follows:

$$\mathcal{L}_\mathcal{C} = -\log \frac{\exp\left(\text{sim}(e_a^{(i)}, e_p^{(i))})/\tau\right)}{\exp\left(\text{sim}(e_a^{(i)}, e_p^{(i))})/\tau\right) + \exp\left(\text{sim}(e_a^{(i)}, e_n^{(i))})/\tau\right)} \tag{3}$$

where $\text{sim}(\cdot, \cdot)$ is the cosine similarity function and $(e_a^{(i)}, e_p^{(i)}, e_n^{(i)})$ is a triplet generated by the mining block, which operates on the batch of embeddings $\mathcal{E}_1$. In particular, $e_a^{(i)}$ (the anchor sample) and $e_p^{(i)}$ (the positive sample) belong to the same subject, while $e_n^{(i)}$ (the negative sample) belongs to a different subject. The hyperparameter $\tau$ (called temperature) controls the magnitude of the similarities.

Finally, our pipeline optimizes the loss defined in Eq. 4, which combines $\mathcal{L}_\mathcal{C}$ and $\mathcal{L}_{\mathcal{BT}}$ by using a weighting function $\beta(t)$ that weights the contribution of each loss at each epoch $t$.

$$\mathcal{L}_{\mathcal{BP}} = \beta(t) \cdot \mathcal{L}_{\mathcal{BT}} + (1 - \beta(t)) \cdot \mathcal{L}_\mathcal{C} \tag{4}$$

The weighting function $\beta(t)$ is inspired by the Profile Weight Functions (Ravi et al., 2022) and is designed to optimize a multi-task learning problem - a combination of two tasks (the self-supervised task and the fully supervised task) - while avoiding training convergence issues. In particular, this function heavily weights the loss from the first task at the start of training and gradually reduces it as training progresses until it is solely focusing on the loss from the second task at the end. In our pipeline, this is achieved using a linear function ($\beta(t) = 1 - \frac{t}{H}$ with H the total number of epochs) which leads to a two-phase training process: the first phase focuses on making the framework invariant to different variability of the scans, while the latter phase focuses on learning contrastive features to distinguish scans of different participants.

### 2.3. Image transformations

In this section, we describe the transformations used in our framework. These transformations (implemented using MONAI (Cardoso et al., 2022)) augment our datasets and generate distorted versions of scans. They are divided into two categories: i) intensity-based –aimed to tackle domain shift variability caused by different contrasts or by different acquisition scanners, and ii) structural based –aimed to make the framework robust to variability related to the subject morphology (e.g., different subject rotations, ageing, disease progression). Our image distortions are obtained by a random sequence of these transformations. More specifically, we define a binary random vector $T = (\phi_1, \ldots, \phi_6)$ where each component $\phi_i$ specifies whether or not the $i$-th transformation –listed in Table 1– is used. Formally, each component $\phi_i$ follows an independent Bernoulli distribution $\phi_i \sim \text{Bern}(p_i)$ with each $p_i$ also specified in Table 1. We define $\mathcal{T} = \text{Bern}(p_1) \times \cdots \times \text{Bern}(p_n)$ as the distribution of the random vector $T$ such that $T \sim \mathcal{T}$.

### 2.4. Default experiment settings

The encoder $\mathcal{R}$ is a ResNet-18 model pre-trained on ImageNet, with the classification layer removed. It generates an output of size $s = 512$. During training, we freeze the first two

PUGLISI[1] INITIATIVE BARKHOF[1,2,3,5] ALEXANDER[1,2] PARKER[1,2,5] ESHAGHI[1,2,5] RAVÌ[1,2,4]

| Transformation | Type | $p_i$ | Parameters |
|---|---|---|---|
| Negative of the image | Intensity-based | 40% | - |
| Intensity shifts | Intensity-based | 40% | sampled in $[-0.25, 0.25]$ |
| Bias field | Intensity-based | 30% | - |
| Rotations | Structural-based | 100% | max 3° |
| Random black patches | Structural-based | 40% | max 3 patch of $10 \times 10$ pixel |
| Elastic deformation | Structural-based | 30% | magnitude range is $[1, 2]$ |

Table 1: Proposed transformations used for image distortion during training.

blocks of convolutional layers and fine-tune the remaining layers. The projection head $P$ is a multilayer perceptron that maps the representations (of size $s = 512$) to an embedding space (of size $z = 2048$), followed by a batch normalization layer and a ReLU activation. After training, the projection head $P$ is discarded and is not used to compute the image fingerprints. The miner adopts the easy positive semi-hard negative strategy described in (Xuan et al., 2020). To train our framework, we use the same configuration settings as (Zbontar et al., 2021). Specifically, we use a LARS optimizer with two different learning rates - one for the weights $(0.2 \cdot b/256)$ and one for the biases $(0.0048 \cdot b/256)$. The training begins with a linear warm-up of 10 epochs, then reduces the learning rate by a factor of 1000 using a cosine decay schedule. Training is performed for 180 epochs with a batch size of $b = 64$. The hyperparameters $\lambda$ and $\tau$ are fixed at 0.0051 and 0.07, respectively. The validation set is used for early stopping. For retrieving similar scans within the dataset consisting of the obtained representations, we used FAISS (Johnson et al., 2019), a fast, parallelized nearest neighbor algorithm that enables sub-linear time search. We measured the mean run-time for extracting the representation of a single scan, which was found to be 3ms, and the run-time for retrieving similar scans for a single query, which was found to be 0.01ms. The workstation specifications used for the training and retrieval are as follows: Intel(R) Core(TM) i3-8100 CPU @ 3.60GHz, RAM 32 GB, Nvidia GeForce GTX 1050 Ti (4GB VRAM).

## 3. Dataset

To validate our framework we used two different datasets. The first consists of 2D slices of skull-stripped T1-MRI scan from ADNI. Here, every subject has on average $10 \pm 7$ scans distributed on an interval of 3 years. The second dataset is obtained synthetically from the first one, by applying different intensity-based transformations to the scans (negative, and random intensity-shift) which have the purpose to simulate the acquisition of different modalities/contrasts of the scans. We will refer to this dataset as SYNT-CONTR. To avoid testing the framework on easy scans that are close in time to the query image (i.e. repeated scans) we sample the datasets such that the follow-up scans of each participant have a distance of at least 180 days. After this data selection, both datasets have 795 scans each (588 used for training, 142 for testing, and 65 for validation).

After the data selection process, both datasets consist of 795 scans from 271 unique subjects. We divided the scans into three sets: 588 for training, 142 for testing, and 65 for validation. These sets are mutually exclusive, meaning that all scans from a single subject belong to only one set, and as a result, there is no overlap between the sets.

## 4. Experimental results

Our experiments are designed to retrieve all the scans of a specific subject given a query image. Inspired by (Musgrave et al., 2020), we used the following protocol to evaluate the performance of our framework: for each scan, in each of the test sets, we retrieve the $k$ most similar representation and we compute two main quality metrics: i) the mean average precision (mAP@R) which takes into account both the precision and the order of the top-k retrieved scans, and ii) the Recall@K (R@K), which indicates the percentage of testing examples whose top-k retrievals include at least one scan from the correct subject. The mathematical definitions of these metrics are reported in Appendix A. We compare our pipeline against state-of-the-art techniques on both the ADNI and SYNT-CONTR test sets. We present these results in Section 4.1 for $K = 3$, while in Section 4.2 we show some visual results of our pipeline. In Appendix B, we also report the result of our approach using different batch sizes (b=64 and b=128) and different formulas for the $\beta(t)$ function to demonstrate its contribution to the training process.

### 4.1. Comparison against state-of-the-art methods

We compare DeepBrainPrint against the following methods: a handcrafted solution that uses the SSIM index (Wang et al., 2004) as a distance metric between scans; a 3D SIFT-Rank brain fingerprinting approach (Chauvin et al., 2020); two Self-Supervised ($\widehat{SS}$) learning techniques (BarlowTwins (Zbontar et al., 2021), SimCLR (Chen et al., 2020)); two standard Fully-Supervised ($\widehat{FS}$) machine learning techniques used for metric learning (NCA (Goldberger et al., 2004) and MLKR (Weinberger and Tesauro, 2007)); and three Fully-Supervised ($\widehat{FS}$) contrastive deep learning techniques (SoftTriple (Qian et al., 2019), InfoNCE (Oord et al., 2018), ProxyNCA (Movshovitz-Attias et al., 2017)) trained using the same backbone architecture employed in our supervised branch. Additionally, where possible, we decide to extend these approaches with our domain-specific Data Transformations ($\widehat{DT}$). The obtained results are reported in Table 2.

Self-supervised methods (BarlowTwins and SimCLR) perform poorly when used with their original image transformations (see Table 2 where $\widehat{DT}$ is not selected). However, their performance improves significantly when we replace their transformations with our proposed ones. On the ADNI dataset, Barlow Twins with our transformations increases the mAP@3 by 199%, and on the SYNT-CONTR dataset by 312%. Similarly, on the ADNI dataset, SimCLR with our transformations increases the mAP@3 by 176%, and on the SYNT-CONTR dataset by 162%. These results show the importance of choosing suitable image transformations for MRI scans and the effectiveness of our proposed ones.

Surprisingly, the SSIM-based solution performs quite well on ADNI (i.e., mAP@3 around 90%). However, its performance drops significantly (from 90% to 49% for mAP@3) when it is applied to SYNT-CONTR. This may be because this approach partially relies on comparing intensity values and fails to capture anatomical structure. NCA and MLKR exhibit similar behaviour, performing well on ADNI but poorly on synthetic data. In this case, their failure may be due to their inability to learn complex structures, even if they are extended using our data transformation technique. We can conclude that SSIM, NCA, and MLKR work well on scans with the same contrast, but are not able to capture the anatomical details of brain structures when applied to images with different contrast.

Puglisi[1] Initiative Barkhof[1,2,3,5] Alexander[1,2] Parker[1,2,5] Eshaghi[1,2,5] Ravì[1,2,4]

| Method | Settings | | | ADNI | | SYNT-CONTR | |
|---|---|---|---|---|---|---|---|
| | $\widehat{\text{FS}}$ | $\widehat{\text{SS}}$ | $\widehat{\text{DT}}$ | R@3 | mAP@3 | R@3 | mAP@3 |
| SSIM-based (Wang et al., 2004) | No training | | | 96.89 | 90.21 | 76.68 | 48.86 |
| 3D SIFT-Rank (Chauvin et al., 2020) | No training | | | **100.00** | **100.00** | 81.77 | 63.71 |
| Barlow Twins (Zbontar et al., 2021) | | ✓ | | 73.06 | 45.35 | 48.70 | 25.52 |
| Barlow Twins with our transformations | | ✓ | ✓ | 97.41 | 90.47 | 92.23 | 79.62 |
| SimCLR (Chen et al., 2020) | | ✓ | | 68.39 | 38.47 | 51.30 | 24.55 |
| SimCLR with our transformations | | ✓ | ✓ | 87.05 | 67.63 | 70.98 | 39.94 |
| NCA (Goldberger et al., 2004) | ✓ | | ✓ | 96.89 | 90.34 | 72.02 | 48.10 |
| MLKR (Weinberger and Tesauro, 2007) | ✓ | | ✓ | 96.37 | 90.03 | 72.02 | 48.07 |
| SoftTriple (Qian et al., 2019) | ✓ | | ✓ | 98.45 | 91.97 | 96.89 | 87.64 |
| Proxy-NCA (Movshovitz-Attias et al., 2017) | ✓ | | ✓ | 98.45 | 90.80 | 94.82 | 84.86 |
| InfoNCE (Oord et al., 2018) | ✓ | | ✓ | 96.89 | 94.04 | 95.34 | 86.95 |
| DeepBrainPrint (Proposed) | ✓ | ✓ | ✓ | 99.48 | 95.54 | **98.96** | **91.00** |

Table 2: Performance (expressed as %) of our framework compared to other methods for the task of patient identification, when evaluated on two different datasets.

Similarly, 3D SIFT-Rank (Chauvin et al., 2020) achieved perfect performance (100% mAP@3) on ADNI compared to our method's (95.6% mAP@3). However, the results from this approach on SYNT-CONTR provide clear indications that keypoint-based methods have domain limitations, including an inability to work correctly on a dataset containing different contrasts/modalities. In particular, our proposed method achieved a +27.3% improvement over (Chauvin et al., 2020) on this second dataset.

The fully-supervised contrastive deep learning methods (SoftTriple, ProxyNCA, InfoNCE) seem to overcome this limitation. They show very high performance on both ADNI (mAP@3 equal to 92%, 91%, and 94%, respectively) and synthetic data (mAP@3 equal to 88%, 85%, and 87%, respectively). Finally, the best performance is achieved by our hybrid method (mAP@3= 96% on ADNI and 91% on SYNT-CONTR). In comparison to the best supervised methods (InfoNCE on ADNI and SoftTriple on SYNT-CONTR), our solution improves mAP@3 by +1.5 and +3.4 percentage points, respectively. In comparison to the best self-supervised method (Barlow Twins with our transformations on both datasets), our solution improves mAP@3 by +5.0 and +11.3 percentage points. This demonstrates the superiority of our approach, which appears to capture the benefits of both self-supervised and supervised techniques. The statistical significance of the obtained improvements was assessed using a paired t-test, with all p-values less than 0.0001.

### 4.2. Visual results

In Figure 2 we show examples of outcomes from our pipeline obtained using six different test images from the ADNI dataset. On the left, we selected three queries where the retrieved scans were all correct. On the right, instead, we selected three queries having a few incorrect results (highlighted in red). As we can see in the latter case, although the retrieved scans belong to subjects with the wrong ID, their brain structures are very similar to the query image. This indicates that our approach can be used to retrieve subjects having similar

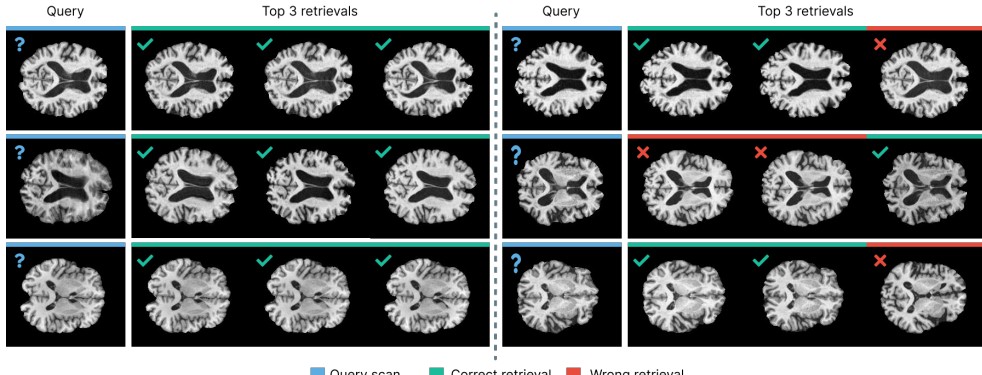

Figure 2: Example of outcomes from our pipeline on six test data extracted from ADNI.

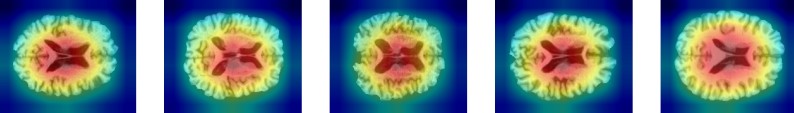

Figure 3: Relevance of the brain regions (saliency map) to the retrieval task.

anatomical details. Finally, in Figure 3 we display the result of an explainable AI framework called Grad-CAM (Selvaraju et al., 2017) which creates a saliency map to visualise which regions of the brain contribute more to the retrieval task. Interestingly, our model seems to focus more on the ventricles and less on the fine detail of the cortex. Although this may seem counterintuitive since central brain structures are shared across humans and many animals, we believe that our approach operated in this way since it uses a hierarchical strategy that places more emphasis on the central brain structures and less on the cortex. This is probably because variations in the central region are more easily discernible and can be used to differentiate between individuals, while the cortex is only examined when the central region is very similar (e.g., individuals from the same family). For more detail on Grad-CAM, please refer to Appendix C.

## 5. Conclusion and future work

In this work, we present DeepBrainPrint, a novel semi-self-supervised pipeline for creating fingerprints of brain MRI scans in real-time and efficiently retrieving images of the same subject within a large dataset. Our results demonstrate that DeepBrainPrint can accurately retrieve previous scans from the same subject, even when the images have different contrast.

Our approach has shown promising results with a recall close to 99% using a single 2D slice, indicating the ability of DeepBrainPrint to capture the similarity of brain structures. To further improve performance, we could try incorporating multiple slices or processing the entire 3D volume, although this may affect real-time processing and resource usage.

In the future, we believe that our pipeline could be useful for various applications, including searching for scans with similar brain shapes, lesions, or atrophy, even if they are not from the same subject. This could be useful for leveraging image-based findings made on previous patients to support diagnostic decisions on new patients or suggest effective treatments for similar disease subtypes/stages.

PUGLISI[1] INITIATIVE BARKHOF[1,2,3,5] ALEXANDER[1,2] PARKER[1,2,5] ESHAGHI[1,2,5] RAVÌ[1,2,4]

## Acknowledgments

Data collection and sharing for this project was funded by the ADNI (National Institutes of Health Grant U01 AG024904) and DOD ADNI (Department of Defense award number W81XWH-12-2-0012). ADNI is funded by the National Institute on Aging, the National Institute of Biomedical Imaging and Bioengineering, and through generous contributions from the following: AbbVie, Alzheimer's Association; Alzheimer's Drug Discovery Foundation; Araclon Biotech; BioClinica, Inc.; Biogen; Bristol-Myers Squibb Company; CereSpir, Inc.; Cogstate; Eisai Inc.; Elan Pharmaceuticals, Inc.; Eli Lilly and Company; EuroImmun; F. Hoffmann-La Roche Ltd and its affiliated company Genentech, Inc.; Fujirebio; GE Healthcare; IXICO Ltd.; Janssen Alzheimer Immunotherapy Research & Development, LLC.; Johnson & Johnson Pharmaceutical Research & Development LLC.; Lumosity; Lundbeck; Merck & Co., Inc.; Meso Scale Diagnostics, LLC.; NeuroRx Research; Neurotrack Technologies; Novartis Pharmaceuticals Corporation; Pfizer Inc.; Piramal Imaging; Servier; Takeda Pharmaceutical Company; and Transition Therapeutics. The Canadian Institutes of Health Research is providing funds to support ADNI clinical sites in Canada. Private sector contributions are facilitated by the Foundation for the National Institutes of Health (www.fnih.org). The grantee organization is the Northern California Institute for Research and Education, and the study is coordinated by the Alzheimer's Therapeutic Research Institute at the University of Southern California. ADNI data are disseminated by the Laboratory for Neuro Imaging at the University of Southern California.

This project has received funding from Innovate UK.

FB, and DCA are supported by the NIHR biomedical research centre at UCLH.

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

## Appendix A. Evaluation metrics

In this section, we describe the quality metrics used in Section 4 to evaluate our pipeline. Let $D$ be the test set. We denote with $x_q^{(i)}$ the $i$-th scan of the test set, and with $x_1^{(i)}, \ldots, x_k^{(i)}$ its $k$-most similar scans retrieved from $D \setminus \{x_q^{(i)}\}$. Also, $y_j^{(i)}$ will represent the patient ID of the scan $x_j^{(i)}$. Let $\mathbb{1}[\cdot]$ be a function that returns 1 if the inner condition is satisfied, and 0 otherwise. The mean average precision (mAP@R) –which is the first quality metric used– is computed as follows:

$$\text{mAP@R}(D) = \frac{1}{|D|} \sum_{i=1}^{|D|} \text{AP@R}(x_q^{(i)}, x_1^{(i)}, \ldots, x_R^{(i)}); \tag{5}$$

where AP@R represents the average precision score:

$$\text{AP@R}(x_q^{(i)}, x_1^{(i)}, \ldots, x_R^{(i)}) = \frac{\sum_{i=1}^{R} \mathbb{1}[y_q^{(i)} = y_i^{(i)}] \left[ \frac{1}{i} \sum_{j=1}^{i} \mathbb{1}[y_q^{(i)} = y_j^{(i)}] \right]}{R}. \tag{6}$$

The mathematical definition of our second quality metric R@K (Recall@$K$) is instead:

$$\text{R@k}(D) = \frac{1}{|D|} \sum_{i=1}^{|D|} \mathbb{1}\left[ y_q^{(i)} \in \{y_1^{(i)}, \ldots, y_k^{(i)}\} \right]. \tag{7}$$

## Appendix B. Pipeline setup

### B.1. Experiment with different weighting functions

In this section, we provide the performances obtained by our pipeline using four different formulations of $\beta(t)$. The first formulation weights the two terms $\mathcal{L}_{\mathcal{NT}}, \mathcal{L}_{\mathcal{BT}}$ of our total loss using a constant function (i.e. $\beta(t) = 0.5$). The second formulation optimizes one term of the loss at a time by using the following step function:

$$\beta(t) = \begin{cases} 1 & \text{if } t < \delta \\ 0 & \text{otherwise} \end{cases} \tag{8}$$

The third formulation is an extension of the previous step function, obtained by iteratively optimising one term at a time on intervals of $\Delta t$ epochs:

$$\beta(t) = \begin{cases} 1 & \text{if } \lfloor t/\Delta t \rfloor \text{ is even} \\ 0 & \text{otherwise} \end{cases} \tag{9}$$

The last formulation weights the terms of the loss using a linear function over training defined as follows:

$$\beta(t) = 1 - \frac{t}{H} \tag{10}$$

where $H$ is the total number of epochs. The hyperparameters of these functions (i.e. $\Delta$ and $\delta$) are optimized using a grid search technique performed on the validation set. The results, reported in Table 3, indicate that the best results are obtained when the linear function is used.

| | | ADNI | | SYNT-CONTR | |
|---|---|---|---|---|---|
| **Weighting function** | **Hyperparameter** | **R@3** | **mAP@3** | **R@3** | **mAP@3** |
| Constant | - | 96.89 | 90.82 | 89.64 | 75.39 |
| Step | $\delta = 30$ | 96.89 | 88.89 | 90.67 | 74.38 |
| Iterative Step | $\Delta t = 30$ | 96.37 | 89.75 | 89.64 | 78.09 |
| Linear | - | **99.48** | **95.54** | **98.96** | **91.00** |

Table 3: Performance of our framework (expressed as %) by using different weighting functions.

### B.2. Experiment with different batch size

We trained our pipeline using two different batch sizes $b = 64$ and $b = 128$ and we found that $b = 64$ provides the best results. These performances are reported in Table 4.

|            | ADNI | | SYNT-CONTR | |
|------------|-------|---------|-------|---------|
| Batch Size | R@3 | mAP@3 | R@3 | mAP@3 |
| 128        | 98.45 | 93.51 | 97.93 | 90.31 |
| 64         | **99.48** | **95.54** | **98.96** | **91.00** |

Table 4: Performance of our framework (expressed as %) by using different batch size.

## Appendix C. Explainability of DeepBrainPrint using Grad-CAM

To obtain the saliency map presented in Figure 3, we computed the Grad-CAM localization map (Selvaraju et al., 2017) for each element of the representation vector calculated from a specific input image and targeting the last convolutional layer of $\mathcal{R}$. We then averaged all of the obtained localization maps (some of which are shown in Figure 4) and normalized them in the range $[0,1]$ to produce the final saliency map. Each point in this final map indicates the importance of the corresponding pixel of the input scan for the retrieval task.

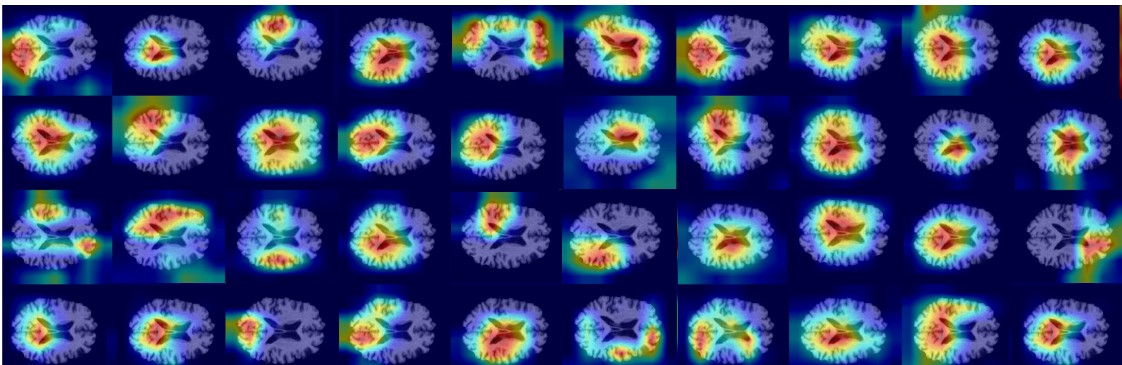

Figure 4: Examples of Grad-CAM activation maps computed on individual elements of the representation vector for a specific input image.

