# OpenReview forum: "DeepBrainPrint: A Novel Contrastive Framework for Brain MRI Re-Identification"
_MIDL.io/2023/Conference — MIDL 2023 Poster_

### Official Review · Reviewer_KQtV · 2023-02-02

**Confidence:** 4
**Preliminary Rating:** 4
**Recommendation:** Oral

**Summary:**

This paper proposes an innovative approach, combining existing methods, for Brain-MRI Re-Identification using a semi-self-supervised contrastive DL approach. The pipeline is divided in two branches (supervised and self-supervised) and includes several transformation as data augmentation strategy. The proposed approach is validated on the ADNI database and synthetic database, where it shows improved performances compared to many other state-of-the-art methods. I believe that both the methods and results would be interesting for the MIDL community.

**Strengths:**

I found that the paper was well written and easy to read. All technical details are specified and the code is available which makes the approach reproducible. The comparison with many existing methods is interesting and shows the benefits of the proposed method.

**Weaknesses:**

A few important information are missing (see comments about section 3 and 4). Some more results could be reported to better compare the tested methods (see comments about section 4; 4.1 and 5) but I understand that space is limited.

**Deanonymize Review:**

no

**Paper Type:**

both

**Questions To Address In The Rebuttal:**

Major:

Section 3 “Dataset”:

•	In the SYNT-CONTR, is the number of scan per subject the same as in the ADNI dataset?

•	I think it would be good to clarify how the training/testing/validation sets were split (within subject scans are not spread between sets)

Section 4 “Experimental results”:

•	I couldn’t find the value of k for the reported results.

•	Also it would be interesting to see the mAP@R and R@K for different k values (e.g., 3 and 6), in Appendix, if space permits. This would show how good the model is at retrieving the furthest scans.

Section 4.1. "Comparison against state-of-the-art methods":

•	Table 2 shows that for the Barlow Twins and SimCLR models, adding the transformations helps a lot. I was wondering if the authors tested their approach without the transformations and how that compares to the other models without transformations (to assess the benefits of the semi-self-supervised approach only). This would be a great addition to the results.

Minor:

Section 5. "Conclusion and future work":

•	This is more of a suggestion for future work but I think it would be very interesting to study the model errors, are the errors correlated with disease progression in the ADNI database (scans from patients with rapid disease progression are harder to retrieve).

---

### Official Review · Reviewer_urN5 · 2023-02-03

**Confidence:** 5
**Preliminary Rating:** 1
**Recommendation:** Poster

**Summary:**

This paper proposes a semi-self-supervised pipeline for creating fingerprints of brain MRI scans in real-time and efficiently retrieving images of the same subject within a large dataset. A fully supervised branch and a self-supervised branch are used,  the self-supervised branch allows the method can learn robustness to synthetic transforms. Results are favorable in the specific context here, however unfortunately are much below existing state of the art for the same task and the same data.

Post rebuttal:
The premise of the original paper was the state-of-the-art performance of their novel network for the brain MRI reidentification task, having relatively high accuracy. However, the paper overlooked the keypoint approach, which achieves perfect accuracy on the same ADNI data and task, including data an order of magnitude larger[1] and multiple real MRI modalities[a]. The authors make efforts to include and understand the keypoint approach, however it appears that a major rewrite is necessary to revise the motivation and situtate this work in the literature, so unfortunately I cannot recommend accept.

Several misunderstandings remain regarding the complexity and performance of algorithms, memory search, particularly as the number of data N grows large. Experimental comparisons, if properly performed, will show the keypoint approach has perfect accuracy for reidentificaiton real or simulated modalities used here. Since the authors approach makes use of 2D images, it would be natural to evaluate this approach also for 2D face reidentification.


**Strengths:**

The paper addresses an interesting topic, re-identification from brain MRI, which may be very relevant for personalized medicine applications and preventing errors.
The idea of incorporating self-supervised learning of transforms is interesting.

**Weaknesses:**

The main weakness in this work is that it is far from state-of-the-art in medical image re-Identification. Specifically for brain MRI, uncited work [1] obtains perfect accuracy for the same task and data, using an order of magnitude more subjects (7500 MRIs including ADNI, OASIS, HCP). The approach here stuggles in top-3 accuracy on a small subset of the data.

Notably [1], requires no training, preprocessing or registration, and is invariant to full 3D similarity transforms (3D scaling, rotation, translation), intensity variations (the approach here considers 2D rotations, no scaling). Numerous instances of mislabelled subjects are identified and verified by dataset administrators, including in ADNI.

[1] Chauvin, L., Kumar, K., Wachinger, C., Vangel, M., de Guise, J., Desrosiers, C., ... & Alzheimer’s Disease Neuroimaging Initiative. (2020). Neuroimage signature from salient keypoints is highly specific to individuals and shared by close relatives. NeuroImage, 204, 116208.


**Deanonymize Review:**

no

**Detailed Comments:**

Recommend for the authors to consider the work [1], including practical aspects as the number of datapoints becomes large.

The paper mentions real-time performance, however no timing is given, nor limitations as to the number of subjects that can be enrolled in training.


**Paper Type:**

both

**Questions To Address In The Rebuttal:**

The main questions would be in comparing to [1].

Were the mislabelled ADNI subjects noticed in this work?

How does network invariance compare to self-supervised learning in the context of reidentification?

Pratically speaking, how does this method generalize as the number of subjects becomes large? Presumably adding new subjects will increase computational requirements and decrease perfomance, but by how much?

How would this method perform in other data contexts, ex. lung CT? Note a method similar to [1] achieved perfect re-identification from 20K lung CT images in IPMI 2015: "A feature-based approach to big data analysis of medical images."

---

### Official Review · Reviewer_afo1 · 2023-02-04

**Confidence:** 4
**Preliminary Rating:** 3
**Recommendation:** Poster

**Summary:**

This paper proposed a representation learning approach for MRI slices and demonstrated its utility for MRI retrieval. The main goal is to learn representations unique to each subject but invariant to age/disease status and other scanner-related differences. Their approach uses two losses --- Barlow twins to impose invariance to image transformations & infoNCE captures unique information about each subject to learn the representations. The proposed method is tested on ADNI and a synthetic dataset created from ADNI to exaggerate contrast variations. The main innovation of the paper is in the MRI-specific image transformation.

**Strengths:**

- The paper does an excellent job of explaining the infoNCE and Barlow twins. However, some details about data and augmentation could be better (See detailed comments).
- The authors compared against several baselines and highlighted traditional baselines' failures in capturing the contrast differences by creating a synthetic dataset. This provided more insights and increased my faith in their method.


**Weaknesses:**

My main complaint with this paper is its poor explanation of supervised/self-supervised terminology. From what I understand, supervised means naturally paired data or scans of the same subject at different times are available, and self-supervised means scans are generated by image transformation. This differentiation appears artificial. Eventually, there is a set of scans belonging to the same subject --- naturally available + generated by augmentation. It would help if the authors could clarify this.

With the above interpretation in mind, I have some specific questions ---

1. Are some of the baselines unfairly crippled? For example, SimCLR can use both observed and transformed scans; but it only uses transformed scans in the SS+DT case.

2. Is the difference between DeepBrainPrint and infoNCE (FS+DT) just the use of Barlow-twin loss and not? If yes, it is interesting to contrast why using Barlow twins in the early part of training improves the result. DeepBrainPrint reduces the contribution of image transformations gradually (because it is used with Barlow-twins), and this begs the question if using the loss function is essential or the decreasing effect of image transformations.

**Deanonymize Review:**

no

**Detailed Comments:**

**Results:** I understand that the choice of k=3 is due to each subject having at least 3 scans. However, this may not be realistic as it is not known apriori how many scans of a subject are in the database. What would the precision and recall look like if the retrieval is done by thresholding similarity?

**Dataset:** How many unique subjects are in the dataset? Sec 3 only mentions the final number of train/test/validation scans. Did you ensure that the subjects did not overlap between the train and test sets?

**Writing:**
- It would be helpful to see more descriptions/mathematical forms of image transformation, especially about the "bias field."
- "The main challenge of this tool is to preserve privacy when searching for data, including avoiding the use of the subject's name." --- This needs to be clarified. How does the tool preserve privacy? Instead, it reveals a loophole in the anonymization and shows that anonymized scans can be almost perfectly identified.
- Use \citet for inline citations. For example, "...Specifically, we extend the approach in (Zbontar et al., 2021)..."


**Paper Type:**

validation/application paper

**Questions To Address In The Rebuttal:**

- Please clarify the questions about supervised/self-supervised terminology (see weakness).
- Please provide details about dataset and image transformations in the paper.
- It would be nice to see the similarity thresholding-based result, but I am generally satisfied with the result presentation.

---

### Meta-Review · Area_Chair_2CKQ · 2023-02-26

**Recommendation:** Accept (Poster)
**Confidence:** 5

**Metareview:**

After reading the extensive debate, it seems that the authors should provide a revision including the Chauvin, L. et al. 2020 NeuroImage as a citation, and the table in [this comment](https://openreview.net/forum?id=i5khDI1te1M&noteId=tM8bNNJx0Hl) should be included in that revision, as part of the appropriate table (currently Table 2). Whether or not these methods run in real time seems immaterial, since no one is proposing to use them in real time.

While the position of the activation on the salience maps challenges current neuro-anatomic beliefs, it is not clear that this is a) in error or b) incorrect for identification via MRI relative to actual biological identification (via dissection or biopsy). Nevertheless, I agree with `urN5` points here, and advise the authors to dig deeper into this issue, as it may either be indicative of idiosyncrasies within their method, or, more optimistically, may have found identifying signal in these regions. I also agree with `urN5` that the T1 and T2 multimodal dataset could also be used from ADNI, and would provide a stronger case for multimodal improvement than the current synthetically generated case.

Otherwise, this manuscript has marginal merit, and I am inclined to accept it conditional on these changes and the PC's discretion.

I have contacted the PC Chair; if the submission receives final acceptance, please include these changes (citation and table addition) in the manuscript.

---

### Meta-Review · Program_Chairs · 2023-02-28

**Recommendation:** Accept (Poster)
**Confidence:** 4

**Metareview:**

After extensive consideration on the part of the program committee, we agree with the area chair that the paper should be conditionally accepted based on the requirements specified in their comment, e.g, inclusion of the updated results and lacking literature.

For future improvements on this paper, I encourage the authors to consider shortcomings, e.g., having less performance than other methods in real data (ADNI), and that the real time execution is not very important for most applications.